# Tidewater-glacier response to supraglacial lake drainage

**Laura A. Stevens** [1] ✉, **Meredith Nettles** [2], **James L. Davis**[2], **Timothy T. Creyts**[2], **Jonathan Kingslake** [2], **Ian J. Hewitt** [3] & **Aaron Stubblefield** [4]

The flow speed of the Greenland Ice Sheet changes dramatically in inland regions when surface meltwater drains to the bed. But ice-sheet discharge to the ocean is dominated by fast-flowing outlet glaciers, where the effect of increasing surface melt on annual discharge is unknown. Observations of a supraglacial lake drainage at Helheim Glacier, and a consequent velocity pulse propagating down-glacier, provide a natural experiment for assessing the impact of changes in injected meltwater, and allow us to interrogate the subglacial hydrological system. We find a highly efficient subglacial drainage system, such that summertime lake drainage has little net effect on ice discharge. Our results question the validity of common remote-sensing approaches for inferring subglacial conditions, knowledge of which is needed for improved projections of sea-level rise.

Fast-flowing (>5 km yr⁻¹) Greenland tidewater glaciers[1–3] contribute to sea-level rise through increased ice discharge[4], sensitively coupled to terminus position[5] and flow resistance at the glacier bed[6,7]. Glacier basal resistance depends on water pressure and distribution within the subglacial drainage system[8,9] and is controlled by where, when, and how much surface melt reaches the bed[10,11]. The flow of the inland Greenland Ice Sheet is sensitive to the variability of surface-melt inputs on hourly to seasonal timescales[5,12,13]. Sea-level-rise contributions from the ice sheet, however, are dominated by the calving of fast-flowing tidewater glaciers at the marine margin[4] —a region where the effects of surface-melt forcing on ice flow are poorly understood[14,15].

Our limited understanding of coupled tidewater-glacier hydrology and ice flow—and the potential future response of tidewater glaciers to expected changes in surface-melt production[16]— is due, in part, to the difficulty of obtaining contemporaneous observations of subglacial water flow and high-temporal-resolution ice-flow velocities for this important class of glaciers. Recent predictions based on inferences from low-time-resolution remote-sensing observations of tidewater-glacier seasonal velocity patterns suggest that changes in flow speeds caused by changes in melt could be very important for ice discharge; or, perhaps, negligible[6,17–19]. These divergent predictions hinge on a widely used binary framework for classifying the subglacial drainage system as efficient, if there is a late-season velocity minimum, or inefficient, if there is not[3,7,12,13,17–20]. This framework is observationally justified for inland ice-sheet regions[12,13], but whether it can be reliably applied to determine the subglacial hydrology of fast-flowing tidewater glaciers is unknown[15]. Determining the nature of the drainage system and its response to meltwater input is critical for sea-level-rise predictions, as increased meltwater could have a tendency to both increase or decrease ice discharge[8–10].

Helheim Glacier (Fig. 1a) is one of the fastest-flowing tidewater glaciers in Greenland[2,21], and has the largest ice discharge at present[4]. Helheim Glacier velocity responds to calving events[21], tides[22], surface melt[23–25], and ice-mélange coherence[2] on sub-seasonal timescales. Unlike many alpine glaciers[11] and inland regions of the Greenland Ice Sheet[12,13], however, Helheim does not exhibit a reduction in velocity over the latter half of the melt season[2,17,18,21], leading some authors to infer that high subglacial water pressures are sustained throughout the melt season[17,18]. Theoretical work[8–10] that matches observations at alpine glaciers[11] would classify such a high-pressure drainage system as inefficient, with surface meltwater input leading to increased basal sliding.

Supraglacial lake drainages[26] offer rare opportunities to observe the ice-flow response to isolated injections of surface melt. Such events have provided key constraints on ice-sheet behavior at lower flow speeds (0.1–0.6 km yr⁻¹)[26–29]. The flow response to lake drainage

¹Department of Earth Sciences, University of Oxford, Oxford, UK. ²Lamont-Doherty Earth Observatory of Columbia University, Palisades, NY, USA. ³Mathematical Institute, University of Oxford, Oxford, UK. ⁴Thayer School of Engineering, Dartmouth College, Hanover, NH, USA. ✉e-mail: laura.stevens@earth.ox.ac.uk

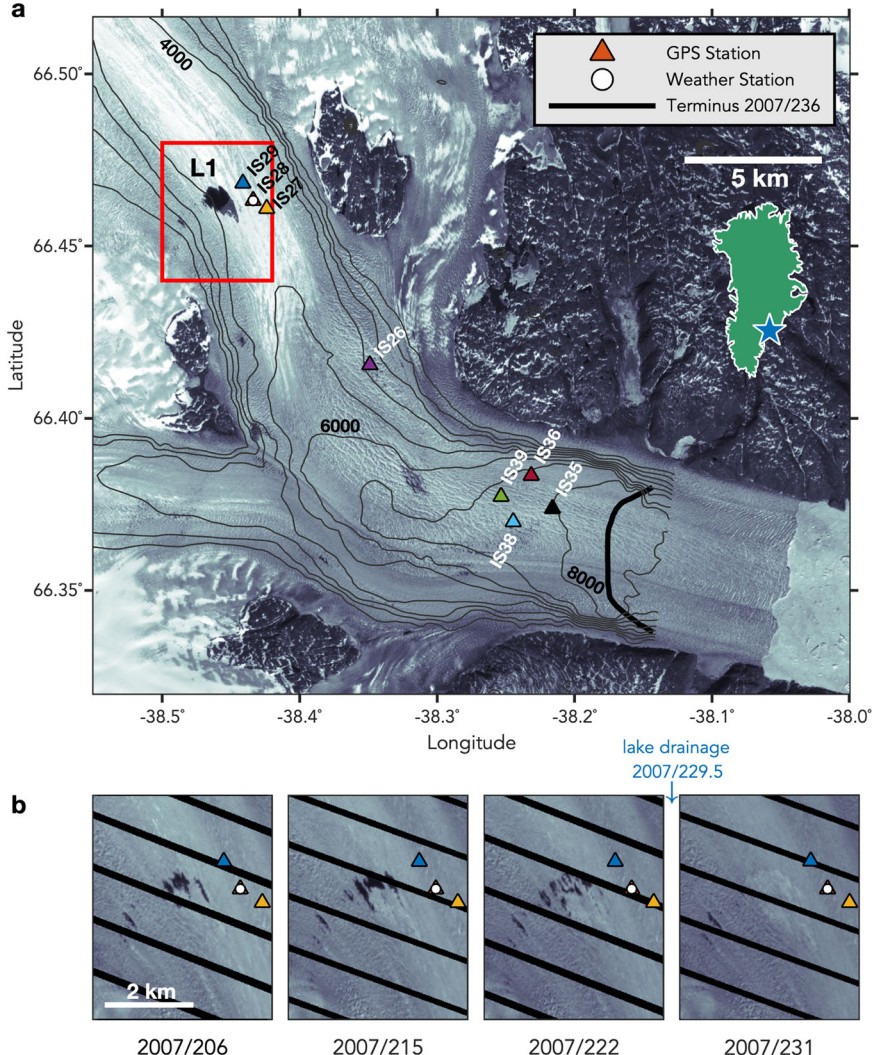

**Fig. 1 | Helheim Glacier, East Greenland. a** Landsat image from 2001/182 showing supraglacial lake L1, (triangles) GPS array, and (circle) Automatic Weather Station (AWS) deployed in 2007. (thick black line) Glacier terminus position on 2007/236. July 2007 surface velocities[64] shown in black contours at 1000 m yr$^{-1}$ intervals. Inset shows (star) location of Helheim Glacier in Greenland. **b** Landsat images of L1 (red box in **a**) from 2007/206–231. Velocity response to L1 drainage begins at 2007/229.5 (Fig. 2). Source data are provided as a source data file. Landsat images courtesy of the U.S. Geological Survey.

along the length of a high-discharge, fast-flowing tidewater glacier has never been measured. Here, we present observations from a lake drainage at Helheim Glacier showing that the hydrological system is dominated by an efficient drainage network that is capable of rapidly exporting the additional meltwater input from the lake, with an increase, and subsequent decrease, in drainage-system water pressure. The water-pressure changes are small and short-lived, such that the lake drainage has minimal impact on the glacier's longer-term average flow speed.

## Results and discussion

Like other major Greenland tidewater glaciers[2,3,6,7], Helheim Glacier hosts multiple supraglacial lakes and water-filled crevasses[30] (Fig. 1a). The lakes fill and drain during the melt season[30,31], exposing optically bright surfaces of smooth, previously submerged ice in the days immediately following drainage[26,30] (Fig. 1b, Supplementary Figs. S1 and S2). In 2007, a network of geodetic-quality, dual-frequency GPS receivers was deployed on the main tributary of Helheim Glacier[21–25] (Fig. 1a; "Methods"). Satellite images on days 222 and 231 of 2007 (2007/222 and 2007/231) (Fig. 1b) and GPS estimates of glacier-surface velocity (Fig. 2a and Supplementary Fig. S3) capture the drainage of supraglacial lake L1 located 18 km from the terminus on 2007/229.

Across all eight GPS stations recording high-quality data (Supplementary Information), the glacier velocity response to lake drainage beginning on 2007/229 is characterized by a 1-day increase in along-flow speed $v$ of ~4%, followed by a reduction in speed to 2% below pre-drainage velocities that is sustained for as much as 2 days (Fig. 2b). We observe ~0.06 m of vertical uplift occurring over 12 h at lake-proximal stations IS27–29 (Supplementary Fig. S4).

The along-flow velocity response to lake drainage begins at lake-proximal station IS27 and reaches near-terminus stations IS35–39 ~3.5 h later (Fig. 2a). We characterize the velocity response with three time points: (1) $t_0$, the time of an initial increase in velocity; (2) $t_{peak}$, the time of maximum velocity response; and, (3) $t_{node}$, the end of a positive velocity response. Each time point occurs first at the stations near the lake, and later at stations farther down-glacier. Stations within <2 km of each other exhibit temporal scatter; time points are within ±0.83 h of each other at lake-proximal stations IS27–29 and ±1.08 h of each other at terminus-proximal stations IS35–39. We interpret $t_0$ to indicate the arrival of high basal water pressures $P_w$ of the subglacial flood[11], and $t_{peak}$ to indicate the time at which the subglacial system becomes maximally over-pressurized[29,32]. At this time, flood waters have likely overwhelmed existing subglacial channels and escaped laterally into neighboring regions of the bed[8,9,13,32,33]. As a network-wide

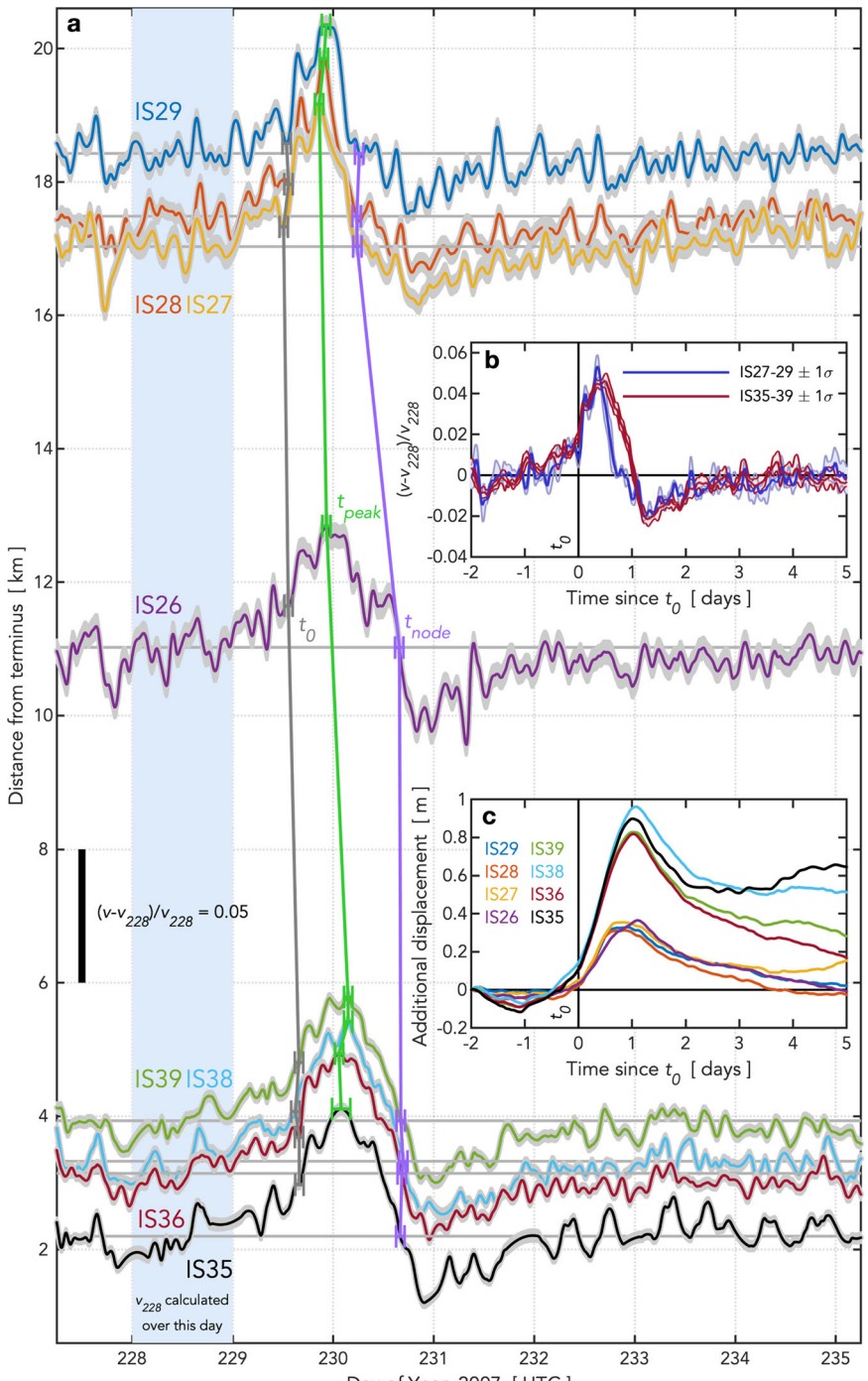

**Fig. 2 | Glacier velocities during supraglacial lake drainage. a** Along-flow velocities $v$ plotted as $(v - v_{228})/v_{228}$, where $v_{228}$ are individual average station velocities on 2007/228. Gray shading shows $\pm 1\sigma$ errors on the velocity ratios. Black bar gives the velocity-ratio scale. Along-flow velocities are plotted by station distance from the glacier terminus; the y-intercept of each horizontal gray line is the station distance from the terminus. Time of (gray) $t_0$, (green) $t_{peak}$, and (purple) $t_{node}$ shown for each station with $\pm 1\sigma$ error bars in time. **b** Averaged velocity pulse ±1 standard deviation of the velocity distribution for stations located near the lake (blue; IS27–29) and terminus (red; IS35–39). Velocity records are temporally aligned at $t_0$. **c** Additional displacement, where a value of 0 m indicates no change in displacement relative to where the station would have been if the station maintained a velocity of $v_{228}$ over the timeseries. Source data are provided as a source data file.

measure of the temporal impact of the lake-drainage event on glacier flow, we use the duration over which the positive velocity response arrives and subsides across the GPS network. Calculated as the time between $t_0$ at IS27 and $t_{node}$ averaged across stations IS35–39, we find a duration of 28.16 h.

Interpreting the glacier velocity pulse and uplift as a response to lake drainage is consistent with additional independent sets of

observations from on- and off-ice Automatic Weather Stations[34] and a pressure transducer record of drainage of the same lake in 2009 that was seen to take 9 h (Supplementary Figs. S2 and S5). Other hypotheses, such as a stationary reduction in basal traction or a terminus-specific forcing, are not compatible with the observations. A stationary perturbation in basal traction cannot explain the velocity pulse because the pulse has a temporal lag as it propagates down-glacier

**Table 1 | Previous observations of down-glacier flood propagation speeds following melt events, rain events, jökulhlaups, and lake drainages**

| Glacier | Propagation speed (m s$^{-1}$) | Glacier type | Event, year | Measurement type | Publication |
|---|---|---|---|---|---|
| Storglaciären, Sweden | 0.01 | Alpine | Diurnal flow over riegel, 1985 | Tiltmeters | Jansson and Hooke[70] |
| LeConte Glacier, Alaska | 0.01 | Tidewater | Rain, 1999 | Optical Survey | O'Neel et al.[71] |
| Franz Josef Glacier, New Zealand | 0.02 | Alpine | Rain, 2011 | GPS | Kehrl et al.[72] |
| Findelengletscher, Switzerland | 0.03 | Alpine | Ice-dammed lake, 1983 | Theodolite | Iken and Bindschadler[11] |
| Unteraargletscher, Switzerland | 0.03 | Alpine | Rain, 1996 | GPS | Gudmundsson et al.[73] |
| Mitdalsbreen, Norway | 0.06 | Alpine | Basal water release, 1987 | Theodolite, Electronic Distance Meter | Willis et al.[74] |
| Black Rapids Glacier, Alaska | 0.07 | Valley | Mini surge, 1987 | Strainmeters | Raymond et al.[75] |
| Variegated Glacier, Alaska | 0.08–0.13 (six events) | Valley | Mini surges, 1980 | Theodolite, Electronic Distance Meter | Kamb and Engelhardt[39] |
| Skaftárjökull, Iceland | 0.1–0.3 | Ice-sheet outlet | Jökulhlaup, August 2008 | GPS | Einarsson et al.[41] |
| White Glacier, Nunavut | 0.17 | Valley | Rain, 1969 | Optical Survey | Iken and Müller[40] Iken[76] |
| Hansbreen, Spitsbergen | 0.17, 0.34 (two events) | Tidewater | Föhn wind melt, 1999 | GPS | Vieli et al.[77] |
| Skaftárjökull, Iceland | 0.2–0.4 | Ice-sheet outlet | Jökulhlaup, 2006 | Not listed | Einarsson et al.[41] |
| Sermeq Avannarleq, West Greenland | ~0.30 | Ice Sheet | Supraglacial lake, 2011 | GPS | Hoffman et al.[38] |
| Kennicott Glacier, Alaska | 0.36 | Valley | Ice-dammed lake, 2006 | GPS | Bartholomaus et al.[68] |
| Skaftárjökull, Iceland | 0.4–0.6 | Ice-sheet outlet | Jökulhlaup, October 2008 | GPS | Einarsson et al.[41] |
| Tungaárjökull, Iceland | ~0.5 | Ice-sheet outlet | Jökulhlaup, 1995 | InSAR | Magnússon et al.[78] as reported in Einarsson et al.[41] |
| Lower Helheim Glacier, East Greenland | 0.52 ± 0.09 | Tidewater | Supraglacial lake, 2007 | GPS | This study |
| Upper Helheim Glacier, East Greenland | 0.96 ± 0.15 | Tidewater | Supraglacial lake, 2007 | GPS | This study |
| Skeiðarárjökull, Iceland | ~1.3 | Ice-sheet outlet | Extraordinary jökulhlaup, 1996 | Flood hydrographs | Björnsson[79] as reported in Einarsson et al.[41] |

over a distance (>16 km) that is 4–8 times larger than the longitudinal stress-coupling length scale[35] of the main tributary of Helheim Glacier[36]. The pulse initiation location and down-glacier propagation direction disqualify a terminus-specific forcing, such as a calving event[21], tidal modulation of flow[22], or a change in mélange coherence[37].

The down-glacier propagation speed of the peak in the velocity pulse we observe at Helheim Glacier represents the fast end-member along a continuum of previously observed pulse-propagation speeds from draining supraglacial lakes[38], ice-dammed lakes[11], stored basal water[39], and precipitation events[40], and is within the range observed for jökulhlaups[41] (Table 1). In the upper terminus region between IS27 and IS26, the peak propagated at $0.96 ± 0.15$ m s$^{-1}$, slowing to $0.52 ± 0.09$ m s$^{-1}$ in the lower-terminus region between IS26 and the four terminus-proximal stations. Here, near the marine terminus, effective pressures $N$ ($N = P_i - P_w$, where $P_i$ is the ice overburden pressure) at the glacier bed have been inferred to be low[24,25]. In general, low effective pressure near the grounding line of marine-terminating outlets promotes subglacial conduit widening, resulting in lower water flow speeds closer to the grounding line[42].

Our observations show that flood events can temporarily modify the speed of even very fast outlet glaciers, but that the net effect on ice advection is small compared to background flow speeds—partly because ice velocity is suppressed below background speeds for ~2 days following the drainage-related peak (Fig. 2a, b). Integrating along-flow velocities from 2 days prior to 5 days following $t_0$ yields an additional, flood-related, ice displacement of −0.03 to +0.15 m for stations IS26–29 and +0.17 to +0.65 m for stations IS35–39 (Fig. 2c). This additional ice displacement during the week of the lake drainage is small relative to background flow speeds of 77–168 m week$^{-1}$ (Supplementary Fig. S3).

The timescale for the drainage system to return to a pre-flood state is longer than the duration of increased velocities, and roughly equivalent to the durations of suppressed velocities observed following mid-melt-season lake drainages on the western margin of the ice sheet[28]. The post-drainage slowdown we observe is consistent with previous observations at other glaciers[11,39] of flow deceleration that follows meltwater supply exceeding a critical rate of water flow[10]. This glacier velocity response, combined with theoretical support from previous numerical modeling[9,10,32,33], indicates that an efficient drainage system, with interacting channelized and cavity components[32,33], exists beneath the glacier at the time of lake drainage.

To test our interpretation that the velocity pulse results from a pressure pulse in the subglacial drainage system, and that a well-developed drainage system is necessary for transporting this pulse down-glacier at the speeds observed in the GPS data, we simulate supraglacial lake drainage in a numerical model[43,44] of subglacial hydrology at Helheim Glacier (Supplementary Fig. S6; "Methods"). The model consists of a continuous subglacial sheet connected to discrete channels at every model node[43]. Following a simulation forced with a fixed basal-melt rate and daily estimates of surface runoff from a regional climate model[45] from 2007/1–229, 0.009 km$^3$ of water is injected to the bed at the L1 location over 9.6 h beginning at 2007/229.5. This set-up and forcing simulates the L1 drainage at the time it occurred in the latter half of the 2007 melt season, when the drainage system has reached an evolved state after ~80 days of meltwater input (Fig. 3a). We refer to this simulation as $M_{229}$ for the day of year corresponding to the modeled lake drainage. Drainage-system response time to runoff input is affected by the choice of values for drainage-system sheet permeability $K_s$ and ice englacial void fraction $\sigma$. We test

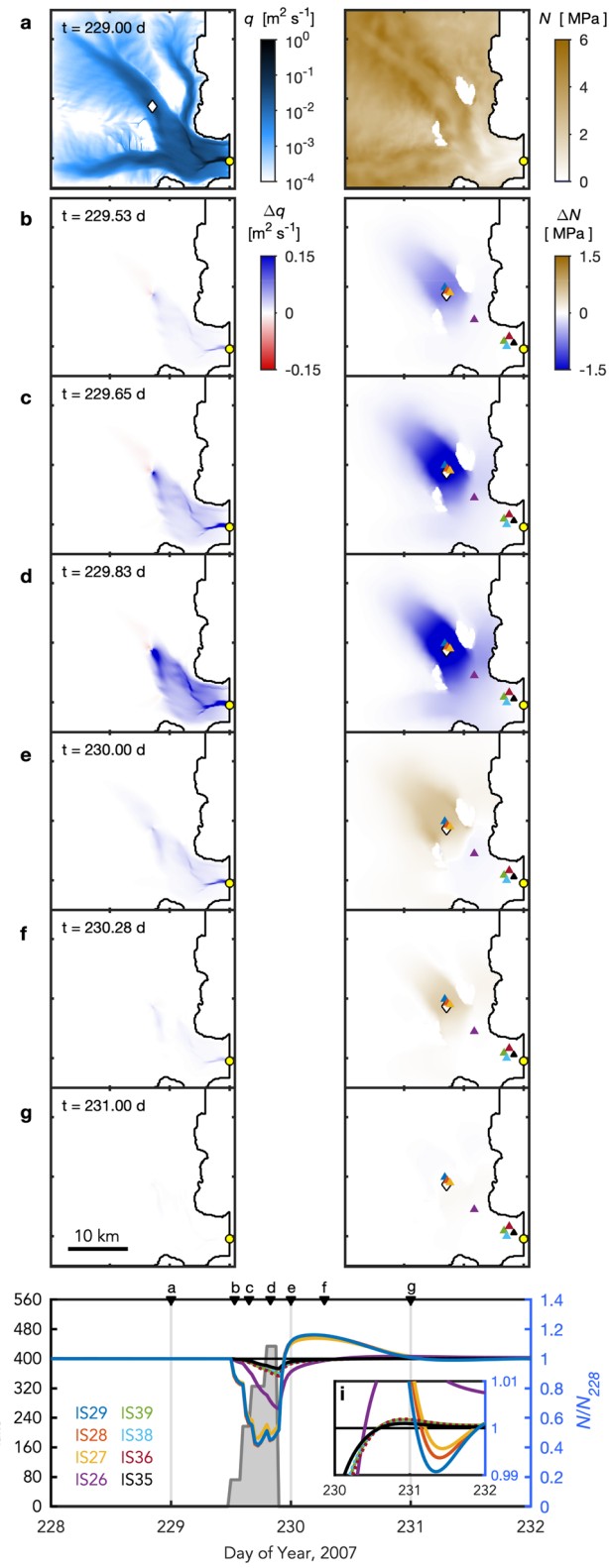

**Fig. 3 | Modeled effective-pressure response to simulated lake drainage. a** (left) Discharge $q$ and (right) effective pressure $N$ prior to simulated rapid lake drainage. The yellow circle shows the discharge outlet location along the glacier terminus. White diamond shows the location of simulated lake drainage. **b–g** Difference between modeled values of $q$ and $N$ at six time points during the simulated lake drainage and the model values shown in **a** at 2007/229.00 and prior to the start of the simulated lake drainage. Triangles show GPS station locations. **h** (gray shading) Prescribed lake discharge $Q_{lake}$ and (curves) modeled effective pressure at the location of each GPS station, plotted as $N/N_{228}$, where $N_{228}$ are individual average values of $N$ at each GPS station location on 2007/228. Black triangles mark time slices shown in (**a–g**). **i** Modeled effective pressure from 2007/230–232 at the location of each GPS station, plotted as $N/N_{228}$, over a finer range in $N/N_{228}$ than shown in (**h**). GPS station colors as in Figs. 1 and 2. Model simulation uses parameter values $K_s = 1\,\mathrm{Pa^{-1}\,s^{-1}}$ and $\sigma = 10^{-6}$. Source data are provided as a source data file.

to higher sliding speeds[9–11,32,33]. Interpreting effective pressure as a proxy for changes in glacier sliding is justified for the purposes of this study because, although the relationship between $N$ and sliding speed can be nonlinear, modeled values of $N$ at the location of each GPS station remain positive during the simulated lake drainage (Fig. 3h), indicating that we have not reached the limiting case where $N$ approaches zero and the relationship between $N$ and sliding speed breaks down[46,47].

Like the positive velocity pulse observed at individual GPS stations, the duration of reduced effective pressure increases from the lake-proximal to terminus-proximal stations (Fig. 3h). In addition, modeled effective pressure shows a post-drainage increase similar in duration to the post-drainage slowdown observed in the GPS data. Modeled effective pressure shows a lower-amplitude response in the near-terminus region than in the region near the lake (Fig. 3), likely due to the lower fractional contribution to water flow from the lake drainage in the near-terminus region—lake, surface runoff, and basal-melt inputs integrate along the bed moving towards the terminus (Fig. 3a)—and because the near-hydrostatic pressure condition at the ocean boundary holds effective pressures closer to flotation in the near-terminus region[24,48] (Fig. 3a). Multiple $M_{229}$ model simulations using a range of reasonable parameter values are able to match GPS pulse durations to within a few hours and to within observed GPS uplift magnitudes (Supplementary Fig. S7). In Fig. 3, we present one of the three best-fitting $M_{229}$ simulations, which reproduces the observed network-wide pulse duration to within 0.7 h and shows 0.02 m of uplift. Thus, our GPS observations and modeling results indicate that an efficient drainage system exists beneath one of Greenland's fastest tidewater glaciers, and that such a system modulates flow while accommodating lake-drainage events.

To verify the importance of the well-developed drainage system predicted by our model in the late melt season for accommodating lake drainage, we create simulations of hypothetical drainages at times when the system is comparatively inefficient[32]. We simulate an equivalent L1 drainage prior to the start of the melt season, when the drainage system receives inputs from basal melt alone and is less developed at the L1 location (Supplementary Fig. S8a). We refer to this simulation as $M_{winter}$ and the simulated L1 drainage begins at 2007/69.5. Though both $M_{229}$ and $M_{winter}$ simulations exhibit broad regions of decreased effective pressure during the drainage events (Fig. 3b–d and Supplementary Fig. S8b–d), the $M_{winter}$ simulation does not exhibit a post-drainage increase in effective pressure and results in a longer pulse duration, with a misfit to our observations of +66.9 h. In agreement with past idealized modeling[33], our results imply that rapid lake drainage into a less-developed drainage system at Helheim, or similar glaciers, would primarily result in reduced effective pressure along the flowline, persisting for a longer time, and would produce net positive glacier advection.

The descriptors efficient and inefficient are often equated with channel-dominated and cavity-dominated drainage systems, respectively[9,10,32,49,50]. Our model allows water to flow through both

a range of values for these model parameters (Supplementary Fig. S7) and consider acceptable values to be those that produce results consistent with the observed GPS velocity-pulse duration and uplift.

Model results show a pulse in reduced effective pressure that transits down-glacier following the onset of the simulated lake drainage (Fig. 3). We present modeled effective pressure $N$ at the location of each GPS station (Fig. 3h), as glacier sliding depends strongly on effective pressure, with low effective pressures thought to correspond

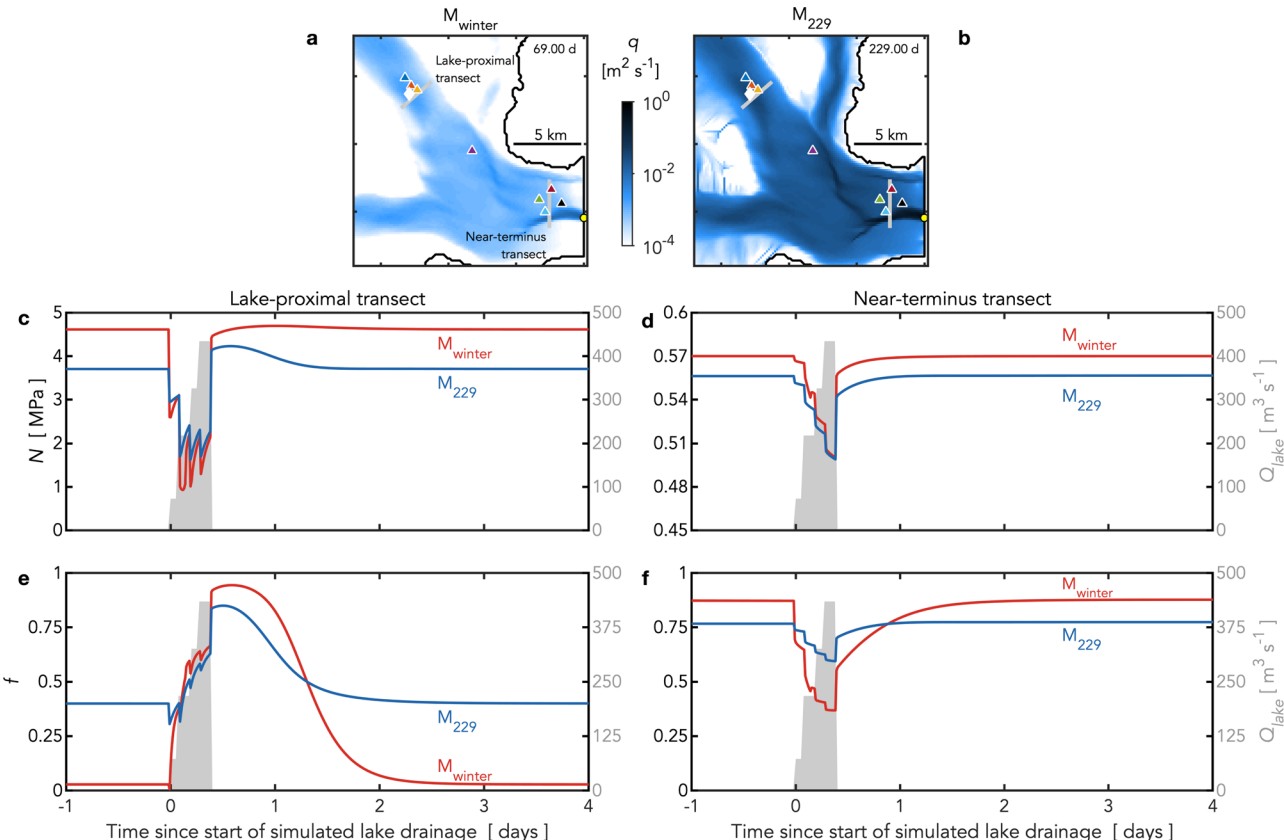

**Fig. 4 | Modeled effective pressure and proportion of discharge carried by channels during simulated lake drainages. a** Discharge $q$ on 2007/69.00 and (gray lines) location of across-flow lake-proximal and near-terminus transects in the model domain. White diamond shows location of simulated lake drainage. Triangles show GPS station locations, with station colors as in Figs. 1–3. **b** Equivalent plot for 2007/229.00. **c** (gray shading; right axis) Prescribed lake discharge $Q_{lake}$ and (left axis) modeled effective pressure $N$ across the lake-proximal transect during (red) $M_{winter}$ and (blue) $M_{229}$ simulated lake drainages. **d** Equivalent plot for the near-terminus transect with different axis limits for modeled effective pressure. **e** (gray shading; right axis) Prescribed lake discharge $Q_{lake}$ and (left axis) the proportion $f$ of modeled discharge carried by channels across the lake-proximal transect during (red) $M_{winter}$ and (blue) $M_{229}$ simulated lake drainages. **f** Equivalent plot for the near-terminus transect. Model simulations use parameter values $K_s = 1\,Pa^{-1}\,s^{-1}$ and $\sigma = 10^{-6}$. Prescribed lake discharge in the $M_{winter}$ and $M_{229}$ simulations begins on 2007/69.50 and 2007/229.50, respectively.

discrete channels and the continuous cavity sheet layer at all points of space and time; the partitioning of the flow between them evolves dynamically over time[43]. We find that flow occurs through a mixture of channels and the sheet layer both during and outside of times of prescribed lake discharge, with spatial variation evident along the glacier (Fig. 4). Near the lake, we observe a decrease in effective pressure and an increase in the flow carried in the channels immediately after the lake discharge begins in both $M_{winter}$ and $M_{229}$ simulations (Fig. 4c, e). Importantly, for the $M_{229}$ simulation, the initial decrease in effective pressure is lower, and is followed by a post-drainage increase in effective pressure (Fig. 4c). By contrast, the $M_{winter}$ simulation shows larger decreases in effective pressure and weak or absent post-drainage increases in effective pressure in both the lake-proximal and near-terminus regions (Fig. 4c, d and Supplementary Fig. S8h). Near the terminus, the flow is dominantly carried by channels, even before lake drainage, in both $M_{winter}$ and $M_{229}$ simulations; the effect of the lake drainage near the terminus is to temporarily reduce the effective pressure by a small amount, and a slightly larger proportion of the water flows in the sheet layer (Fig. 4e, f).

Together, these results indicate that, for Helheim Glacier, an efficient drainage system cannot simply be equated with a channel-dominated system. The system is able to rapidly adjust to the extra lake water input, even though a significant proportion of the pre-lake-drainage flow is carried by the cavity sheet (Fig. 4e, f). Our observations (Fig. 2) and model results (Figs. 3 and 4) suggest a better

test for efficient drainage beneath glaciers like Helheim Glacier—lightly grounded[2], fast-flowing tidewater glaciers with high meltwater throughput—would be whether increased meltwater input leads to net neutral or reduced ice advection.

At Helheim, when lakes drain during the melt season[30], the effect of a single lake-drainage event on ice advection is small compared to background flow speeds. With a daily, lower-terminus ablation rate of ~0.04 m d⁻¹ water equivalent (w.e.) over a four-month melt season[34], the basal drainage system receives high meltwater volumes from abundant surface crevasses[30]. This setting facilitates a well-developed drainage system that allows rapid response to and recovery from a sudden lake-drainage event. The relative insensitivity of net glacier advection to lake drainage observed at Helheim, however, is unlikely to extend to tidewater glaciers that experience lower seasonal meltwater forcing. For example, remote-sensing observations show that sporadic austral summer-melt events (<0.005 m d⁻¹ w.e. for 1-week duration) at multiple Antarctic Peninsula tidewater glaciers coincide with ice-flow accelerations of up to 100% above annual speeds[20]. These flow accelerations are not followed by sustained periods of reduced velocities and result in net positive glacier advection, likely because water pressure spikes into an inefficient drainage system[20], consistent with our model results for simulated winter-season lake drainage at Helheim Glacier. Thus, while these Antarctic Peninsula[20] and other, currently colder, High Arctic tidewater glaciers[51] may initially undergo a net speed-up as the amount of surface meltwater reaching the bed increases, their flow response to individual melt events may eventually

decrease as higher seasonal melt magnitudes are attained as the climate continues to warm[52].

Observational and theoretical advances are needed along the surface-melt continuum to test this response-evolution hypothesis and improve projections of global tidewater-glacier response[15,16] to expected changes in surface-melt production on timescales longer than individual melt or lake-drainage events. In particular, we have argued that the Helheim Glacier flow response to lake drainage provides strong evidence that the lake drains into an efficient drainage system. Helheim Glacier's near-terminus seasonal velocity pattern has, by contrast, been invoked to infer that an inefficient drainage system persists throughout the melt season[18]. Our observations suggest that near-terminus seasonal velocity patterns at glaciers of this type may not be well correlated with drainage-system efficiency, or the corresponding capacity to accommodate meltwater input with minimal impact on glacier discharge. At fast-flowing tidewater glaciers, where a marine margin and high surface melt rates keep subglacial water pressure high[24], observations of seasonal velocity patterns[17–19] may be misinterpreted to indicate inefficient drainage, where, in fact, an efficient system exists and is capable of exporting large volumes of meltwater rapidly. The widely used binary framework for interpreting subglacial drainage system efficiency from seasonal velocity patterns[3,7,17–20], though observationally justified for inland ice-sheet regions[12,13], does not explain the full range of behaviors observed at lightly grounded[2], fast-flowing tidewater glaciers with high meltwater throughput.

## Methods

### GPS data
A network of geodetic-quality, dual-frequency GPS receivers was deployed from late June to late August 2007. During the time of the L1 lake drainage, the network consisted of eleven receivers that spanned an along-flow distance of 2–24 km from the calving front. Additional fixed stations were located at bedrock sites. GPS data were processed in kinematic mode using the TRACK software package[53] to yield position estimates every 15 s[21–23]. Here, we eliminate position estimates with unfixed biases, and rotate the timeseries to obtain position estimates in the local along-flow and vertical directions at each station. We use a stochastic filter to estimate a horizontal along-flow velocity $v$, a principally semi-diurnal horizontal ocean-tide response, and a diurnal component of horizontal glacier flow[23,24]. Step changes in along-flow velocity are allowed at times of glacial earthquakes, which represent large calving events[21,23,24] (Supplementary Fig. 3). Eight of the eleven receivers in operation recorded data of sufficient quality for high-resolution, stochastic-filter analysis (Supplementary Methods). We focus our analysis on the along-flow velocity $v$, which is largely free of glacier flow responses to tidal[22] and diurnal[23,24] forcing (Supplementary Figs. 9–17).

We characterize the velocity response to lake drainage using three time points: (1) $t_0$, the time of an initial increase in velocity; (2) $t_{peak}$, the time of maximum velocity response; and, (3) $t_{node}$, the end of the positive velocity response (Supplementary Fig. 3). The time of $t_0$ is picked by eye to be immediately prior to the acceleration in $v$ beginning after 2007/229.5. The time of $t_{peak}$ is the time of the maximum value of $v$ after $t_0$. The time of $t_{node}$ is the time when $v$ first crosses back below $v_{228}$, the average value of $v$ on 2007/228 at each station, following $t_{peak}$. To facilitate interstation comparison of the ice-flow response to lake drainage, we calculate the fractional change in $v$ over the lake-drainage event at each station. We present the ratio of $(v - v_{228})/v_{228}$ in Fig. 2a. Values of $v_{228}$ range from 11–24 m d$^{-1}$, with larger $v_{228}$ observed at near-terminus stations (Supplementary Fig. 3). To produce the averaged velocity-pulse curves in Fig. 2b, the $(v - v_{228})/v_{228}$ ratios are aligned in time based on the pulse onset time $t_0$ at each station, and then averaged across stations within 2 km of each other.

We calculate the speed of the down-glacier propagation of the velocity pulse by differencing $t_{peak}$ between adjacent stations and then dividing by interstation distance measured along the flowline. Uncertainties in this estimate are calculated using a ±0.02 m uncertainty in GPS station position equivalent to 1σ uncertainty in horizontal position associated with the TRACK position solutions, and an uncertainty in $t_{peak}$ estimated at ±1 h for all stations except IS35. A data gap at station IS35 during the drainage event requires $v$ to be interpolated across the time of $t_{peak}$, resulting in an uncertainty in $t_{peak}$ of ±2.16 h for this station (Supplementary Methods; Fig. 2a and Supplementary Fig. S17).

### Additional observations supporting the velocity-pulse interpretation
In addition to the primary datasets of the 2007 GPS data and satellite imagery, four independent sets of observations are consistent with and further support the interpretation of the glacier velocity pulse as a response to the rapid drainage of lake L1. First, contemporaneous temperature and net short-wave radiative flux observations at an on-ice Automatic Weather Station (AWS) show no anomalously high values in the days leading up to the drainage, suggesting that the velocity pulse is not related to a period of above-average surface melt (Supplementary Fig. 5). Second, contemporaneous observations at the Tasiilaq off-ice AWS are consistent with those from the glacier, and show no anomalous temperature, insolation, relative humidity, or precipitation in the days prior to the event. The 6-hr precipitation observations in Tasiilaq record a rain event on 2007/231–233; however, this event occurs after elevated along-flow velocities have subsided, suggesting that the velocity pulse is not due to a rain event. Third, an L1 rapid drainage was recorded in 2009 by a water-pressure transducer deployed in the lake (Supplementary Fig. 2). Though no GPS velocities are available during the 2009 event, the pressure transducer records a 9-hr drainage duration, providing evidence that the lake can drain rapidly, with a duration similar to that of other rapid supraglacial lake drainages in Greenland[26,27,54,55]. Finally, we observe no change in other supraglacial water bodies or the structure of the proglacial ice mélange between sequential satellite images spanning the event (Supplementary Fig. 1). The velocity pulse is thus not likely due to the drainage of a supraglacial water body other than L1 or to a change in back pressure on the glacier terminus. We do observe changes in water levels of ice-dammed lakes along the eastern side of the northern tributary of Helheim Glacier at the time of the L1 drainage (Supplementary Fig. 1); however, the subglacial drainage pathways for these ice-dammed lake drainages would not transit beneath stations IS27–29 based on the subglacial topography (Supplementary Fig. 6a) and modeled discharge routes (Fig. 3a) for the northern tributary.

### Automatic weather station data
We use AWS data recorded on[34] and off[56] the glacier to evaluate meteorological conditions during the time of lake drainage. The on-ice AWS was located near IS28 (66.46˚N, 38.44˚W) (Fig. 1a), and recorded a standard suite of meteorological parameters at an hourly sampling rate, including temperature, relative humidity, and incoming and reflected short-wave radiative fluxes (Supplementary Fig. 5). The net short-wave radiative flux (insolation) is the AWS variable most closely correlated with the total energy flux available for melting the glacier surface[34]. We compare the on-ice AWS observations to an off-ice AWS located ~102 km to the southeast in Tasiilaq[56] (65.60˚N, 37.62˚W). The off-ice AWS recorded hourly temperature and relative humidity and a record of precipitation every 6 h.

### Water-pressure-transducer data
An internally logging HOBO water-pressure transducer was deployed within the L1 lake basin (66.46˚N, 38.46˚W) from late June to late August 2009, and recorded measurements of water depth at an hourly sampling rate (Supplementary Fig. 2). From 2009/205–234, the raw

water-level record shows five, 0.5–2.5 m step offsets that we attribute to instrument malfunction. We estimate the magnitude of each step offset by eye and shift water-depth measurements up after each offset such that measurements are continuous with the relatively cleaner record from 2009/192–204. We then use a moving-average filter 0.1-d in width to identify and remove observations more than 3σ from the mean to produce the relative water-depth record shown in Supplementary Fig. 2.

### Subglacial-hydrology model
We simulate supraglacial lake drainage in a numerical model of subglacial hydrology at Helheim Glacier. We use a two-dimensional model of subglacial drainage[43] most recently employed to investigate seasonal drainage-system dynamics of land- and marine-terminating regions of the western Greenland Ice Sheet margin[44,57]. Full model equations are given in ref. 44.

The model, which is similar to the Glacier Drainage System model[50], routes surface meltwater input into a continuous sheet connected to discrete channels melted upward into the base of the ice sheet[43]. Water moves between a continuous sheet, channels, and englacial storage to maintain a continuous hydraulic potential. The continuous sheet has a thickness $h$, which is the sum of a cavity sheet layer with thickness $h_{cav}$ and an elastic sheet layer with thickness $h_{el}$, which is included to represent elastic uplift of the glacier when $N$ becomes negative ($P_w > P_i$). The thickness of the cavity sheet evolves due to the combined effects of basal ice melt, cavity opening by basal sliding, and cavity closing by ice creep[58,59]. Water flux through the sheet $\mathbf{q_s}$ is dependent on the coefficient $K_s$ controlling the sheet permeability, and sheet thickness $h$. Though $K_s$ is a spatially uniform parameter, the effective hydraulic transmissivity of the sheet layer $K_s h^3$ varies in both space and time in response to water input.

Water flux in the sheet $\mathbf{q_s}$ is connected to discrete channels with discharge $q_Q$ at every model node. The growth and decay of channel cross-sectional area is a competition between melt opening and creep closure of channel walls[60]. The incipient sheet width contributing to channel melting $\lambda_c$ sets the length scale over which ice melting contributes to channel formation. Mass conservation is a balance between the sheet, channel, and englacial storage components. Englacial storage is dependent on the connected void fraction of the ice $\sigma$ and the cross-sectional area of moulins $A_m$ (Supplementary Table 1).

For comparison with the horizontal and vertical GPS data, we focus our analysis on model predictions of $N$, total flux $q$ (the combined flux from the channel and sheet layers), and sheet thickness $h$, which are solved at every model grid point spaced at 150 m. The ice-flow model in ref. 43 is not used here, as the exact form of the sliding law relating $N$ to basal sliding velocities is uncertain, can be highly nonlinear, and would introduce an additional set of unknown model parameters. Instead, we interpret modeled $N$ as a proxy for changes in glacier velocity during simulated supraglacial lake drainages. The use of the ref. 43 hydrology model alone prevents a direct coupling between basal sliding and rates of cavity opening, which can result in a negative feedback when additional cavity space is opened during times of faster sliding[61]. However, due to the small magnitude (<5%; Fig. 2b), short-duration (1-day) velocity changes we observe in response to the L1 drainage, we expect increases in basal sliding to have a minor impact on rates of cavity opening.

### Model domain and boundary conditions.
The model domain is an ~800 km² region that extends over the three main tributary branches of Helheim Glacier (Supplementary Fig. 6), and consists of a regularly spaced, rectangular mesh of model nodes with 150-m spacing. We use the 150-m resolution IceBridge BedMachine Greenland v3 bedmap[62] to represent bedrock topography, and the 150-m resolution Greenland Ice Mapping Project digital elevation model[63] to represent glacier-surface elevation. The basal sliding speed $U_b$ at each node is set to the value of the corresponding surface speed from the MEaSUREs Greenland Ice Sheet Velocity compilation[64,65] for July 2007. Bedrock topography, glacier-surface elevation, and basal sliding speed are assumed to be constant.

The model is forced by a constant basal-melt rate and daily surface runoff. The basal-melt rate $m$ is prescribed everywhere to be 0.0262 m yr⁻¹ based on an average geothermal heat flux of 0.063 W m⁻² beneath Greenland[66] and a contribution to basal melt from frictional heating due to sliding estimated using a basal shear stress $\tau_b$ of 60 kPa and a basal sliding velocity of 500 m yr⁻¹. Surface runoff is derived from downscaled 1-km-resolution Regional Atmospheric Climate Model v2.3 (refs. 45, 67) runoff estimates, and interpolated to each of the model grid nodes[44]. Because surface meltwater enters the basal drainage system through abundant crevasses on Helheim Glacier[30,34], we input surface runoff to the glacier bed at every model node. There is no surface storage term in the model, and surface runoff transits immediately to the bed. For boundary conditions, a small upstream basal-melt flux is applied at the eastern boundary on the subglacial drainage system, equivalent to the basal-melt rate (0.0262 m yr⁻¹; Supplementary Table 1) integrated over a 50-km-distance inland from the eastern boundary. We prescribe $N$ to be zero at both marine- and land-terminating margins of the glacier. This boundary condition represents the approximately hydrostatic pressure expected at the marine terminus, given observations of glacier surface height and bed elevation that indicate the glacier is near flotation on 2007/236[24] and interpretations of a transiently floating terminus in more recent years[31] when the terminus has retreated back to near its location on 2007/236 (Fig. 1a).

### Simulated supraglacial lake drainage.
Daily changes in runoff forcing prevent the model from reaching a steady state during the melt season[43,44]. In order to isolate the effect of a supraglacial lake-drainage event on the drainage system, we first force the model with daily runoff from January 1, 2007 to December 31, 2007. We then explore the effect of L1 drainages by beginning the lake-drainage events on different days of the year. For $M_{229}$ scenarios, we initiate the lake drainage at approximately the same time as $t_0$ inferred from the GPS data: 2007/229.50. This date falls late in the melt season, when the model receives inputs from both surface runoff and basal melt. For $M_{winter}$ scenarios, we initiate the lake drainage on 2007/69.50, when the drainage system receives inputs from basal melt, but surface-melt input is negligible. At 2007/69.00, discharge is largely accommodated in the sheet layer in the lake-proximal region (IS27–29) (Supplementary Fig. S8a). Thus, initiating a lake drainage on 2007/69.50 simulates injecting a lake into an inefficient drainage system at the L1 location.

The simulated lake-drainage volume, injection location, and duration is equivalent for $M_{229}$ and $M_{winter}$ scenarios. Because the exact drainage volume and duration are unknown for the 2007/229 L1 drainage, these values are approximated based on observations of L1 drainage duration in 2009 (Supplementary Fig. 2) and L1 maximum volume in 2007 (ref. 30). The volume estimate given in ref. 30 was obtained from a digital elevation model of the post-drainage L1 basin on 2007/205. Our lake-drainage simulation injects 0.009 km³ of water in total over 9.6 h and across four neighboring model nodes located within the extent of L1 on the glacier surface. Prescribed lake discharge $Q_{lake}$ increases in a step-wise fashion over the 9.6 h to a maximum of 434 m³ s⁻¹ (Fig. 3h), which is within the range of discharge rates for rapidly draining supraglacial lakes[26,54]. The full model run is 7 days in length, extending in time from 2007/228–235 for $M_{229}$ scenarios and 2007/68–75 for $M_{winter}$ scenarios. Basal-melt and surface-runoff inputs across the entire model domain are held fixed during the simulated lake-drainage event at 2007/227 values of 0.0004 km³ d⁻¹ and 0.0147 km³ d⁻¹, respectively, summed across the model domain for $M_{229}$ scenarios and at 2007/67 values of 0.0004 km³ d⁻¹ and 0 km³ d⁻¹, respectively, summed across the model domain for $M_{winter}$ scenarios.

This is done to isolate the effective-pressure response to the lake-drainage event from daily changes in effective pressure at day boundaries due to daily changes in surface-runoff forcing.

**Model parameter-space calibration.** Model parameter values are chosen from previous work applying this model to the western margin of the Greenland Ice Sheet[44,57] (Supplementary Table 1). The bed roughness height scale $h_r$ and length scale $l_r$ are equivalent to those used in refs. 44, 57 (Supplementary Table 1). The primary justifications for using these values are that they keep us in a previously examined region of this model's parameter space[44,57] and that they are similar to length scales for bed roughness used by previous studies[10,43,68]. We vary sheet permeability $K_s$ by factors of 10 across the range $10^{-4}$–$10^{1}\,Pa^{-1}\,s^{-1}$, and englacial void fraction $\sigma$ by factors of 10 across the range $10^{-8}$–$10^{-2}$. This results in 42 different parameter combinations (Supplementary Fig. 7). The model converged for all 42 parameter combinations. At high englacial void fractions, the volume of englacial storage space (0.01 km$^3$) is greater than the volume of water in the simulated lake drainage (0.009 km$^3$), which results in lake-drainage inputs being mostly accommodated within the englacial void space. In these cases, a response in $N$ is observed at the lake-proximal stations (IS27–29), but no response is observed at the terminus-proximal stations (IS35–39) by the end of the model run, which extends 5.5 days after the onset of lake drainage.

We assess parameter-space fitness using two criteria. First, we compare the observed pulse duration for the GPS network to modeled values of $N$ at the location of the GPS stations in the model domain (Supplementary Fig. 7a, b). The pulse duration in the GPS data is calculated as the time between $t_0$ at station IS27 and $t_{node}$ averaged across stations IS35–39 to be 28.16 h. The pulse duration in the model is calculated as the time between $t_0$ of $N$ at station IS27 and $t_{node}$ of $N$ averaged across stations IS35–39, where $t_0$ of $N$ is the time when $N/N_{228} < 0.98$ and $t_{node}$ of $N$ is the time when $N/N_{228} > 1.00$ following $t_{peak}$ (Fig. 3h). We calculate the misfit between the observed and modeled pulse duration by differencing the two values, and present the misfit in hours (Supplementary Fig. 7a, b).

Second, we compare the amount of observed vertical uplift at stations IS27–29 following $t_0$ to modeled changes in drainage-system height (Supplementary Fig. 7c, d). In the vertical GPS data, we observe ice-sheet surface uplift of 0.06 m over 12 h beginning at 2007/229.5 at stations IS27–29 (Supplementary Fig. 4). We do not observe similar uplift at other GPS stations. The 0.06 m of uplift observed at stations IS27–29 is equivalent to 2 standard deviations in uncertainty in vertical height associated with the TRACK position solutions, and we would have observed larger vertical uplifts had they occurred. We, therefore, exclude models where a change in the height of the sheet layer $h$ during the course of the simulated lake drainage exceeds 0.09 m, equivalent to 3 standard deviations in vertical position uncertainty, at the location of stations IS27–29. This uplift criterion is satisfied for 24 (57%) of the scenarios (Supplementary Fig. 7c, d). When $K_s \leq 10^{-2}\,Pa^{-1}\,s^{-1}$ and $\sigma \leq 10^{-3}$, changes in $h$ during the simulated drainage range from 0.15 to 0.86 m, and thus do not satisfy the uplift criterion.

Overall, both pulse duration and uplift are more strongly controlled by sheet permeability than by englacial void fraction. The pulse-duration criterion provides a finer model-space calibration than the uplift criterion. One of the three best-fitting $M_{229}$ simulations has a pulse-duration misfit of −0.7 h (42 min), with $K_s = 1\,Pa^{-1}\,s^{-1}$ and $\sigma = 10^{-6}$ (Supplementary Fig. 7a). Model outputs from this $M_{229}$ scenario and an $M_{winter}$ scenario with equivalent parameter values are shown in Fig. 3 and Supplementary Fig. 8, respectively.

## Data availability
Landsat images are available from the United States Geological Survey (https://www.usgs.gov/). GPS data are archived at UNAVCO (www.unavco.org/data). The on-ice AWS data are archived at the Geological Survey of Denmark and Greenland (GEUS) (https://doi.org/10.22008/FK2/LDEMCY). The off-ice AWS data[56] are archived at the Danish Meteorological Institute (http://research.dmi.dk/data/). Glacier-surface elevation is from the Greenland Ice Mapping Project digital elevation model[63] archived at the National Snow and Ice Data Center (NSIDC) (https://nsidc.org/data). Bedrock topography is from the IceBridge BedMachine Greenland v3 bedmap[62] archived at the NSIDC (https://nsidc.org/data/IDBMG4). Glacier-surface speed in July 2007 is from the MEaSUREs Greenland Ice Sheet Velocity compilation[64,65] archived at the NSIDC (https://nsidc.org/data). Source Data underlying figures in this study are archived at https://doi.org/10.5281/zenodo.7023662[69].

## Code availability
The subglacial-hydrology model code used for this study is archived at https://doi.org/10.5281/zenodo.7023662[69].

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

## Acknowledgements

Data collection was supported primarily by the Gary Comer Science and Education Foundation, U.S. National Science Foundation (NSF) award 0713970, the Danish Commission for Scientific Investigations in Greenland (KVUG), and the Spanish Ministry of Science and Innovation. We thank members of the 2007 Helheim Project for collecting the GPS and AWS data used in this study and acknowledge in particular the contributions of A.P. Ahlstrøm, M.L. Andersen, P. Elosegui, G.S. Hamilton, T.B. Larsen, and L.A. Stearns to the fieldwork and earlier analysis of the data. We are grateful to B.P.Y. Noël and M.R. van den Broeke for the RACMO output. Support to L.A.S. was provided by a Lamont-Doherty Earth Observatory Postdoctoral Fellowship and a John Fell Oxford University Press Fund. M.N. and J.K. were supported by the NSF award 2003464. T.T.C. was supported by a Vetlesen Foundation grant, NSF award 1643970, and NASA award NNX16AJ95G. GPS equipment and technical support were provided by UNAVCO, Inc.

## Author contributions

All authors conceived the study. M.N. and members of the Helheim Project carried out the fieldwork. J.L.D. developed the stochastic filter. L.A.S., M.N., and J.L.D. performed the stochastic analysis. I.J.H. developed the subglacial model. L.A.S. performed the model simulations. L.A.S., M.N., J.L.D., T.T.C., J.K., I.J.H., and A.S. interpreted the results. L.A.S. made the figures and wrote the paper. M.N., J.L.D., T.T.C., J.K., I.J.H., and A.S. commented on the paper.

## Competing interests

The authors declare no competing interests.
