## [Peer Review File · Nature Communications]

Tidewater-glacier response to supraglacial lake drainageREVIEWER COMMENTS

Reviewer #1 (Remarks to the Author):

This paper presents older data from Helheim Glacier showing the impact of a lake drainage event on ice flow speeds/discharge. The paper is quite well-written and the data is thoughtfully presented in a few figures (plus supplementary information). The main point of this paper is that the overall impact of a single lake drainage is more or less negligible for Helheim implying that the subglacial system accepts the additional water easily – and is thus an ‘efficient’ drainage system. The paper demonstrates this nicely with GPS data as well as a model showing both the reproduced lake drainage event as well as the impact of a lake drainage during more ‘winter-like’ conditions. The latter experiment has a stronger impact on ice discharge as expected because the subglacial system is less efficient in winter. The conclusions are thus that interpretation of seasonal velocity changes for outlet glaciers cannot be a simple interpretation of runoff-controlled changes in drainage system efficiency – because many melt-dominant outlet glaciers maintain efficient conditions all summer. I think that this result is noteworthy as it calls into question how glaciologists interpret velocity changes more broadly. Outlet glaciers are controlled by myriad processes and this paper goes one step further to clarifying those controls than previous attempts.

My sense is that this paper has seen many prior reviews since the work is so well-presented (or perhaps the authors are just ‘that good!’). As a result, I only have minor changes to suggest as follows.

1. Could you say any more about the lake volume change over time? Looks like there may have been some volume reduction between 215 and 222. Also, it seems that the timing of lake drainage is only from GPS – there are no other data confirming that 229 was the day of lake drainage? If so, then Line 73-75 should be a more nuanced statement that the satellite observations merely bracket the GPS-determined time of lake drainage – but do not confirm that it drained on 229.

2. Line 78: I think referencing Fig. 2b is more appropriate here

3. I was all ready to cry foul on the interpretation of your results because lake drainages would have a different impact when they were less-melt dominated and then the authors went ahead and proved that! This is a strong result that demonstrates that nuanced interpretation of melt-induced velocity changes is required.

4. Line 114: the first mention of a model is in this line with very little introduction – I know space is tight and the model methods are explained elsewhere but a few words on what the goal of the model would be worth a mention – ah, I see it more adequately introduced on Line 133 – I think some wordsmithing is simply needed here

5. I found the methods confusing with the mention of 2009 data ahead of the section titled “Additional observations supporting the velocity pulse interpretation” I would suggest this be the leading section of the Methods.

6. Supplementary: Can you continue GPS station colour coordination through the SI as well?

Reviewer #2 (Remarks to the Author):

Review "Tidewater-glacier response to supraglacial lake drainage" by Stevens and others, submitted to Nature Communication

By analyzing a dataset from 2007 showing a lake drainage at Helheim glacier, the authors can conclude that the state of the basal drainage system at the time of this lake drainage (summer) is highly efficient such that the lake drainage has little effect on the mean ice discharge. The paper is well written and easy to follow, the figures are of good quality.

My main issue is about the fact that the hydrology model is used alone, without any coupling with

an ice flow model. Indeed, as there is a direct coupling between basal sliding velocity and the opening of the basal cavities, how much these potential negative feedback would change the results? It should be at least discussed. Also, it would allow a direct comparison with the observed velocity, which is crucially missing here as the comparison is only done using the effective pressure. Depending on the state of the hydrological system, the relation between sliding velocity and effective can be highly non linear, and for the limit case where N tends to zero (Coulomb type regime), there is even no relation at all between these two variables. How much the effective pressure proxy from the model can be really used alone to explain the observed surface velocity from the GPS? This should be at least discussed.

Other comments:

- line 15: why do you state that the increasing surface melt on discharge is unknown for tidewater glaciers, in opposition to land-terminated glacier? My understanding is that the processes are the same, but that the effect of an increase of melt water is more visible on land-terminated glaciers than on tidewater ones? But our knowledge on this process is as well (por badly) known for both?

- line 37: one the same line, I would said that this is difficult to observe even for land-terminated glaciers. Don't understand what make the specificity of the tidewater glacier. All these arguments look more like a nice storyboard for a paper than a real differences.

- line 48: things are evolving dynamically, may be add "until it evolves to an efficient drainage system and velocity slow down"?

- line 54: high water pressure or a sufficient amount of basal melt produced such that the drainage system stay efficient throughout the melt season? (because high water pressure would mean the drainage system stay inefficient throughout the melt season).

- line 74: I found it is missing some numbers about the total volume of water estimated in the lake, compared to the total volume of water available for the basin (daily and total melt of the summer), to have an idea of the order of magnitude of these different terms at play.

Reviewer #3 (Remarks to the Author):

Review of "Tidewater-glacier response to supraglacial lake drainage"

by Laura A. Stevens et al.

Submitted to Nature Communications

Overview

This manuscript describes observations of velocity response following a supraglacial lake drainage event on Helheim Glacier in east Greenland in 2007, then uses a subglacial hydrology model to simulate the lake drainage and infer qualities of the subglacial drainage system. The main finding is that Helheim likely has a highly efficient subglacial drainage system capable of rapidly accommodating large volumes of water, in contrast to the near-terminus seasonal velocity pattern that is typically associated with inefficient drainage systems. This is important, as it highlights the fact that characterizing subglacial drainage based on near-terminus surface velocity seasonal patterns may be inaccurate in fast-moving tidewater glaciers.

The paper is clearly written, nicely organized, and well referenced. This work brings together numerical modeling and field observations in a helpful way. I am not an expert in GPS deployment and stochastic filtering methods, so I cannot comment critically on those aspects of this paper, but the described methods sound reasonable.

A few main points:

The authors caution against the binary approach commonly used for classifying subglacial drainage based on near-terminus seasonal velocity patterns. I am, however, a bit dismayed that the premise of the paper rests firmly on another binary distinction between "inefficient" and "efficient"

subglacial drainage systems. This way of thinking about subglacial hydrology distinguishes between different drainage components and applies different equations to each (notably, only allowing for opening by melt in channel components). It is a nice model, but it may be worthwhile to acknowledge this classification and consider how it may be limited in its representation of the full spectrum of drainage types that exist in the natural system. Especially in glaciers with complex topography like Helheim where we might expect to see a lot of spatial heterogeneity, it seems beneficial to move toward a more nuanced treatment of the drainage than to label the entire subglacial system as being "efficient" or "inefficient". This isn't to say that the main finding of the paper that centers around finding an "efficient" drainage system beneath Helheim is faulty, as it nods to the established classification system based on pressure response to meltwater input; I would simply recommend adding some discussion around this theme. I also suggest noting that the sheet permeability and englacial void ratio, invoked in the model as uniform, constant values over the domain, likely vary spatially and temporally in reality (also mentioned in the specific comments below).

Finally, I realize this is beyond the scope of this paper, but it would be interesting to explore the same modeling problem with an ice-sheet model coupled to the subglacial-hydrology model, allowing sliding velocity to evolve rather than using modeled changes in effective pressure as a proxy for velocity changes. As the authors state (lines 371-374), this would introduce a host of other uncertain parameters associated with the sliding relation. The authors should be well-equipped to explore these nonlinear feedbacks in further work, and I encourage them to do so.

Overall, I enjoyed reading this manuscript. With some small revisions, I feel it will make a worthy contribution to Nature Communications. In addition to the points mentioned above, please find specific comments and questions below.

Specific comments

Figure 2: The along-flow velocities are plotted as $(v-v_{228})/v_{228}$ as stated. I am confused about the horizontal gray lines, however: the caption states that these are where $(v-v_{228})/v_{228}=0$ for each individual station. But if the left axis is showing $(v-v_{228})/v_{228}$ for each station, wouldn't the horizontal gray lines all be 0? Perhaps I am missing something. I am not sure I understand the stacking of the velocity profiles; I understand that they are stacked on the right axis based on distance from terminus, but the velocity values on the left axis are perplexing.

Figure 3: It is interesting to see the upstream effects of the lake drainage, particularly lower effective pressure (i.e. higher water pressure), and lower flux (in b, c, and d). This upstream behavior doesn't show up in the Mwinter simulation – this could be worth commenting on and discussing more. The caption is slightly confusing as to what differences are being plotted. I think it is the difference between each time slice and the initial configuration shown in a prior to the lake drainage, correct?

Figure 3: Only the stations near the lake drainage appear to experience the increase in N following the lake drainage (on day 230) that is visible upstream and downstream of the lake location. Can this be explained in more detail? Are the stations near the terminus far enough removed that the pulse is dampened by the time it reaches them? Is the drainage system not "efficient" near the terminus? Is there sufficient water flowing in from the other branch to mute the influence of the lake drainage?

Figure 3: In panel h, it is difficult to distinguish all the different station lines, as some are stacked. It looks like only IS27, 28, and 29 experience the positive rebound in N after the conclusion of the lake drainage, which is the whole premise of this paper. So, what's going on at the bed around the lake compared to near the terminus? More detailed explanation or interpretation would be helpful.

Line 386: It would be helpful to describe what the basal melt rate represents and how it is calculated. (Geothermal flux and frictional heat from sliding? Although that should vary spatially, and it sounds like it is a uniform constant rate here). Is this basal melt applied everywhere in the model, or only to channel components?

Line 395: I am not convinced that $N=0$ is necessarily appropriate for the terminus. I believe Helheim is generally thought to be grounded at the terminus, but $N=0$ implies a floating terminus. It would be helpful to explain why this choice was made, as opposed to setting the subglacial water pressure at the discharge equal to the overlying water pressure in the fjord.

Lines 428-468: Permeability and englacial void ratio are prescribed as uniform over the entire domain. Finding the best fit values as done here may be helpful as a homogenized representation of what's happening at the bed, but may be misleading by blurring the spatial heterogeneity.

Supplementary Figure 4: For how many days after the drainage event does the vertical lowering continue in IS 27-29?

Supplementary Table 1: It would be helpful to explain the choices for b_r and l_r used in the model.

Response to reviewers of *Tidewater-glacier response to supraglacial lake drainage* by Stevens et al. submitted to *Nature Communications*

Reviewer #1 (Remarks to the Author):

This paper presents older data from Helheim Glacier showing the impact of a lake drainage event on ice flow speeds/discharge. The paper is quite well-written and the data is thoughtfully presented in a few figures (plus supplementary information). The main point of this paper is that the overall impact of a single lake drainage is more or less negligible for Helheim implying that the subglacial system accepts the additional water easily – and is thus an ‘efficient’ drainage system. The paper demonstrates this nicely with GPS data as well as a model showing both the reproduced lake drainage event as well as the impact of a lake drainage during more ‘winter-like’ conditions. The latter experiment has a stronger impact on ice discharge as expected because the subglacial system is less efficient in winter. The conclusions are thus that interpretation of seasonal velocity changes for outlet glaciers cannot be a simple interpretation of runoff-controlled changes in drainage system efficiency – because many melt-dominant outlet glaciers maintain efficient conditions all summer. I think that this result is noteworthy as it calls into question how glaciologists interpret velocity changes more broadly. Outlet glaciers are controlled by myriad processes and this paper goes one step further to clarifying those controls than previous attempts.

My sense is that this paper has seen many prior reviews since the work is so well-presented (or perhaps the authors are just ‘that good’!). As a result, I only have minor changes to suggest as follows.

Thank you for your review. This is the first round of peer review for the manuscript.

1. Could you say any more about the lake volume change over time? Looks like there may have been some volume reduction between 215 and 222. Also, it seems that the timing of lake drainage is only from GPS – there are no other data confirming that 229 was the day of lake drainage? If so, then Line 73-75 should be a more nuanced statement that the satellite observations merely bracket the GPS-determined time of lake drainage – but do not confirm that it drained on 229.

Yes, in 2007, the satellite images show the lake surface area to vary over the melt season including between 2007/215 and 2007/222. The reviewer is also correct that, for the lake drainage on 2007/229, the timing of the lake drainage is derived from satellite and GPS data. We infer the timing of the lake drainage from the horizontal-velocity response captured by the GPS observations, and this horizontal-velocity response is bracketed by the available satellite images, which show a nearly complete loss of lake area from 2007/222 to 2007/231. We revised the sentence on L73 to better describe the above.

2. Line 78: I think referencing Fig. 2b is more appropriate here

Reference changed to Fig. 2b.

3. I was all ready to cry foul on the interpretation of your results because lake drainages would have a different impact when they were less-melt dominated and then the authors went ahead and proved that! This is a strong result that demonstrates that nuanced interpretation of melt-induced velocity changes is required.

Thank you.

4. Line 114: the first mention of a model is in this line with very little introduction – I know space is tight and the model methods are explained elsewhere but a few words on what the goal of the model would be worth a mention – ah, I see it more adequately introduced on Line 133 – I think some wordsmithing is simply needed here

Thank you for pointing this out. We have removed the clause “consistent with our model estimates of N (Fig. 3a)” from this sentence ending on L114, as we agree with you that mentioning the model here, without a proper introduction, is confusing.

5. I found the methods confusing with the mention of 2009 data ahead of the section titled “Additional observations supporting the velocity pulse interpretation” I would suggest this be the leading section of the Methods.

We have revised the section order such that **Additional observations supporting the velocity pulse interpretation** is now the second section in the Methods, following the **GPS data** section. We believe the GPS methods should come first, as they constitute the primary observations used to analyze the lake-drainage event in the main-text figures; and the **GPS data** section does not refer to the 2009 data.

6. Supplementary: Can you continue GPS station colour coordination through the SI as well?

We have changed the colors in Supplementary Figures 3 and 4 to make the GPS station color consistent throughout all figures in the manuscript and SI. Supplementary Figures 9–17 remain in black and white.

Reviewer #2 (Remarks to the Author):

By analyzing a dataset from 2007 showing a lake drainage at Helheim glacier, the authors can conclude that the state of the basal drainage system at the time of this lake drainage (summer) is highly efficient such that the lake drainage has little effect on the mean ice discharge. The paper is well written and easy to follow, the figures are of good quality.

My main issue is about the fact that the hydrology model is used alone, without any coupling with an ice flow model. Indeed, as there is a direct coupling between basal sliding velocity and the opening of the basal cavities, how much these potential negative feedback would change the results? It should be at least discussed. Also, it would allow a direct comparison with the observed velocity, which is crucially missing here as the comparison is only done using the effective pressure. Depending on the state of the hydrological system, the relation between sliding velocity and effective can be highly non linear, and for the limit case where N tends to zero (Coulomb type regime), there is even no relation at all between these two variables. How much the effective pressure proxy from the model can be really used alone to explain the observed surface velocity from the GPS? This should be at least discussed.

Thank you for your review. We agree that, ideally, the hydrology model would be coupled to a calculation of ice-flow velocity to directly test the relationship between sliding velocity and subglacial hydrology. As noted by Reviewer 3, such a model is beyond the scope of this paper due to the additional set of uncertain parameters that would be added if a sliding relation were included. While the *Hewitt* (2013) hydrology model has been successfully coupled to ice-flow for idealized geometries, the coupled hydrology-ice-flow model has not been extended to realistic topographies. The main difficulty in developing the coupled hydrology-ice-flow model to realistic topographies is that obtaining reasonably realistic velocity fields requires inverting for a spatially variable basal drag. How to complete that inversion properly is the topic of ongoing research. We note that previous studies using coupled models have had to employ less sophisticated hydrological models. We do intend to explore such model developments in future work.

However, due to the small magnitude of the velocity changes we observe during the lake-drainage event (less than $\pm 5\%$; Fig. 2b), we would not expect the opening of basal cavities during the lake-drainage event to have a substantial negative feedback on the ice velocities. We observe mean flow speeds U_b at the stations of $10\text{--}25\text{ m d}^{-1}$ (Supplementary Figs. 3, 6c); adjusting sliding velocity by $+5\%$ for one day during the lake-drainage event would have a minor effect on rates of cavity opening. The effect is likely to be much larger for inland Greenland Ice Sheet regions (*Joughin et al., 2008; Pimentel and Flowers, 2010; Hewitt, 2013*) where observations show greater variability in diurnal-velocity amplitudes ($\sim 100\%$ above mean speeds), lake-drainage velocity amplitudes ($\sim 200\%$ above pre-drainage speeds), and melt-season seasonal-velocity amplitudes ($\sim 50\%$ above winter speeds). We have added an explanation to this effect at the end of the Methods section **Subglacial hydrology model**.

We agree with your point that the relation between sliding velocity and effective pressure depends on the value of effective pressure and that, as effective pressure approaches zero, the relationship between sliding velocity and effective pressure breaks down. It is important to note that the bed beneath the near-terminus GPS stations IS35–39 has a modelled effective pressure of $0.4\text{--}0.9\text{ MPa}$ on the day before the start of the simulated lake-drainage event for both M_{229} and M_{winter} scenarios. The bed beneath these stations returns to an effective pressure of $0.4\text{--}0.9\text{ MPa}$ following the pulse of decreased effective pressures moving through. Moreover, as values of N/N_{228} (Fig. 3h) and N/N_{68} (Supplementary Fig. 8h) do not dip below zero for any GPS-station locations for either lake-drainage simulation—indicating that N remains above zero in these locations during the lake-drainage simulation—we are not close to the limiting case where N is zero. We have added a sentence explaining the above on L154–158.

Other comments:

- line 15: why do you state that the increasing surface melt on discharge is unknown for tidewater glaciers, in opposition to land-terminated glacier? My understanding is that the processes are the same,

but that the effect of an increase of melt water is more visible on land-terminated glaciers than on tidewater ones? But our knowledge on this process is as well (por badly) known for both?

We have revised this sentence on L15 to clarify that the effect of increasing surface melt on *annual* rates of discharge from tidewater glaciers is unknown. While the effect of increasing amounts of annual melt in inland regions of the Greenland Ice Sheet is generally thought to result in slower annual ice velocities (e.g., *Tedstone et al., 2015*), a similar, observationally supported direct relationship between runoff and ice-flow speeds—one that does not conflate the additional runoff-driven processes of terminus melt and retreat—has not been parsed for Greenland outlet glaciers (e.g., *Catania et al., (2020), Section 5.3, Atmospheric Controls on Outlet Glaciers*).

- line 37: one the same line, I would said that this is difficult to observe even for land-terminated glaciers. Don't understand what make the specificity of the tidewater glacier. All these arguments look more like a nice storyboard for a paper than a real differences.

Contemporaneous on-ice GPS observations of surface velocities and observations of proglacial river discharge in land-terminating regions of the Greenland Ice Sheet have been completed by many researchers (e.g., *Cowton et al. (2013)*). Such contemporaneous observations of on-ice GPS observations of surface velocities and full-season, subglacial discharge locations and rates across submerged tidewater glacier termini and have yet to be completed. The difficulty of obtaining such measurements in a tidewater-glacier setting remains one of the major observational hurdles in the field of ice-ocean interactions. For a review discussion on this for Greenland tidewater glaciers, see *Catania et al. (2020), Section 5.3, Atmospheric Controls on Outlet Glaciers*, specifically: "...there is currently no way to validate the time-varying discharge of subglacial water that runs off the GrIS into the ocean. Research addressing this question for subglacial runoff in land-terminating areas suggests variable model success in capturing observations."

- line 48: things are evolving dynamically, may be add "until it evolves to an efficient drainage system and velocity slow down"?

We have revised the ending to this paragraph in response to this comment, and the first major comment from Reviewer 3. Sentence now reads: "Determining the nature of the drainage system and its response to meltwater input is critical for sea-level rise predictions, as increased meltwater could have a tendency to both increase of decrease ice discharge^{8–10}" (L45–47).

- line 54: high water pressure or a sufficient amount of basal melt produced such that the drainage system stay efficient throughout the melt season? (because high water pressure would mean the drainage system stay inefficient throughout the melt season).

We are not sure of the reviewer's intent in this comment, as we have not discussed basal melt within the Introduction. In the final two sentences of this paragraph (L52–56), we mean high subglacial water pressures (L52–54), and that theoretical work would suggest that high-pressure drainage systems indicate an inefficient drainage system (L55–56).

- line 74: I found it is missing some numbers about the total volume of water estimated in the lake, compared to the total volume of water available for the basin (daily and total melt of the summer), to have an idea of the order of magnitude of these different terms at play.

The maximum volume of water that could potentially drain from L1 (0.009 km³) is given on L140 and in the Methods subsection *Simulated supraglacial lake drainage*. We have added an additional sentence to the Methods subsection giving the amount of daily surface runoff input to the entire model domain during the M₂₂₉ lake-drainage simulations (0.0147 km³ d⁻¹) and the amount of water sourced from basal melt input (0.0004 km³ d⁻¹).

Reviewer #3 (Remarks to the Author):

Overview

This manuscript describes observations of velocity response following a supraglacial lake drainage event on Helheim Glacier in east Greenland in 2007, then uses a subglacial hydrology model to simulate the lake drainage and infer qualities of the subglacial drainage system. The main finding is that Helheim likely has a highly efficient subglacial drainage system capable of rapidly accommodating large volumes of water, in contrast to the near-terminus seasonal velocity pattern that is typically associated with inefficient drainage systems. This is important, as it highlights the fact that characterizing subglacial drainage based on near-terminus surface velocity seasonal patterns may be inaccurate in fast-moving tidewater glaciers.

The paper is clearly written, nicely organized, and well referenced. This work brings together numerical modeling and field observations in a helpful way. I am not an expert in GPS deployment and stochastic filtering methods, so I cannot comment critically on those aspects of this paper, but the described methods sound reasonable.

A few main points:

The authors caution against the binary approach commonly used for classifying subglacial drainage based on near-terminus seasonal velocity patterns. I am, however, a bit dismayed that the premise of the paper rests firmly on another binary distinction between “inefficient” and “efficient” subglacial drainage systems. This way of thinking about subglacial hydrology distinguishes between different drainage components and applies different equations to each (notably, only allowing for opening by melt in channel components). It is a nice model, but it may be worthwhile to acknowledge this classification and consider how it may be limited in its representation of the full spectrum of drainage types that exist in the natural system. Especially in glaciers with complex topography like Helheim where we might expect to see a lot of spatial heterogeneity, it seems beneficial to move toward a more nuanced treatment of the drainage than to label the entire subglacial system as being “efficient” or “inefficient”. This isn’t to say that the main finding of the paper that centers around finding an “efficient” drainage system beneath Helheim is faulty, as it nods to the established classification system based on pressure response to meltwater input; I would simply recommend adding some discussion around this theme. I also suggest noting that the sheet permeability and englacial void ratio, invoked in the model as uniform, constant values over the domain, likely vary spatially and temporally in reality (also mentioned in the specific comments below).

Thank you for this comment, which has led us to think more carefully about how we could best communicate the distinction between “inefficient” and “efficient” drainage within our study. First, we now identify explicitly in the manuscript the common practice of equating “efficient” with “channelized” drainage, and “inefficient” with distributed or sheet/cavity drainage. Second, we have added two new paragraphs and a new figure (Figure 4) to the Results section to illustrate how the drainage system evolves with time and space. The glacier system is “efficient” by the measure that it evacuates a slug of water quickly, and there is little net additional displacement due to this extra water.

Our model uses the hydrology components of *Hewitt* (2013) over a realistic bed topography for Helheim Glacier. In this model, flow is always allowed to flow through both discrete channels and the continuous cavity sheet layer at any point and at any time. In fact, we find that total water flow is always some mixture of the two. In some cases (or at some times), the majority of the water is carried by the cavities, and in other cases (or at other times), the majority of the water is carried by channels. It is important to note that the hydrological dynamics themselves *choose* this distribution of water flow; the distribution of water flow is not imposed. (Though the dynamics are, of course, influenced by choices in model parameters including englacial void fraction and sheet permeability.) It is also important to note that, although the reviewer is correct that only melt opening is allowed in the channel components, in every location in the model domain the drainage system is made up of both cavities and channels, so that

opening in the drainage system is allowed by both melt-opening growth in the channel components *and* an increase in the height of the cavity sheet layer. Given this, the *Hewitt* (2013) model as it is employed in this study describes the spectrum of possible subglacial drainage types between “inefficient” and “efficient” drainage in a more gradational way than the reviewer has initially interpreted.

The reviewer’s interpretation is, probably in large part, due to our lack of discussion of these points in the main text of the study. We have added two new paragraphs and a new figure (Figure 4) to the end of the Results section to try to remedy this. Figure 4 shows the proportion of subglacial water flow that is routed through channels during the M_{229} and M_{winter} simulations across two transects: one transect immediately down-flow from the lake-proximal GPS stations and a second transect running through the cluster of the four near-terminus GPS stations. Among other details, this figure shows (1) that water flow occurs through a mixture of channels and the sheet layer in times of both zero (M_{winter}) and late-season (M_{229}) runoff forcing, and (2) that the definition of an “efficient” drainage system need not rest on the assumption that all, or nearly all, of water flow occurs through channels. Please see Figure 4 and the new paragraphs in the Results section for the full details (L188–210).

We have chosen not to introduce our study using the interpretations and language we arrive at by the end of our study. In some instances in the Introduction section, we rely on established definitions for classifying subglacial drainage systems, but we try to leave the door open for our departure from these classifications later on. At the end of the revised Results section, we now call out the standard assumptions, and suggest that a better test for efficient drainage, at least for Helheim Glacier and perhaps other tidewater glaciers, would be if increased runoff forcing leads to neutral or reduced ice advection. As the reviewer noted, this definition for “efficient” aligns with the established classification system based on the pressure response to meltwater inputs (*Schoof*, 2010) and our modeled effective-pressure response to the simulated lake drainages (Fig. 4c,d).

Please see our response to your comment on model sheet permeability and englacial void ratio in response to your line comment below for L428-468.

Finally, I realize this is beyond the scope of this paper, but it would be interesting to explore the same modeling problem with an ice-sheet model coupled to the subglacial-hydrology model, allowing sliding velocity to evolve rather than using modeled changes in effective pressure as a proxy for velocity changes. As the authors state (lines 371-374), this would introduce a host of other uncertain parameters associated with the sliding relation. The authors should be well-equipped to explore these nonlinear feedbacks in further work, and I encourage them to do so.

We agree that developing an appropriate coupling between the hydrology model and an ice-flow model for realistic geometries of Helheim Glacier would be a great avenue for future work. Please see our response above to Reviewer 2’s main comment, where we discuss our reasoning for not pursuing a coupled hydrology-ice-flow model in this current study.

Overall, I enjoyed reading this manuscript. With some small revisions, I feel it will make a worthy contribution to Nature Communications. In addition to the points mentioned above, please find specific comments and questions below.

Thank you for your review.

Specific comments

Figure 2: The along-flow velocities are plotted as $(v-v_{228})/v_{228}$ as stated. I am confused about the horizontal gray lines, however: the caption states that these are where $(v-v_{228})/v_{228}=0$ for each individual station. But if the left axis is showing $(v-v_{228})/v_{228}$ for each station, wouldn’t the horizontal gray lines all be 0? Perhaps I am missing something. I am not sure I understand the stacking of the velocity profiles; I understand that they are stacked on the right axis based on distance from terminus, but the velocity values on the left axis are perplexing.

We have edited Figure 2 and the figure caption to make it clearer. We had previously added the velocity ratios $[(v-v_{228})/v_{228}]$ to station distance from the terminus (in kilometers) to make the station distance from the terminus (on the former right y-axis) equivalent to where $[(v-v_{228})/v_{228} = 0]$. However, our previous wording in the caption, and the inclusion of increasing values on the along-flow velocity y-axis, was confusing. We have now removed the vertical axis with increasing along-flow velocity values in favor of just giving a vertical scale bar that indicates the amplitude of the velocity ratio. We have revised the sentence in the caption where the horizontal grey line is defined to read: “Along-flow velocities plotted by station distance from the glacier terminus; the y-intercept of each horizontal grey line is the station distance from the terminus.”

Figure 3: It is interesting to see the upstream effects of the lake drainage, particularly lower effective pressure (i.e. Higher water pressure), and lower flux (in b, c, and d). This upstream behavior doesn't show up in the M_{winter} simulation – this could be worth commenting on and discussing more. The caption is slightly confusing as to what differences are being plotted. I think it is the difference between each time slice and the initial configuration shown in a prior to the lake drainage, correct?

The reviewer is correct that the values shown in Figure 3 **b–g** are the model output at the time noted on the left panel less the model output at $t=299.00$ d shown in panel **a**. We have revised the sentence in the caption that explains these panels to make this more clear: “**(b–g)** Difference between modeled values of q and N at six time points during the simulated lake drainage and the model values shown in **a** at 2007/229.00 prior to the start of the simulated lake drainage.” We have also revised the equivalent sentence in the Supplementary Figure 8 caption.

While discharge upstream of the simulated lake-drainage location does decrease more, at least in an absolute sense, in the M_{229} simulation than in the M_{winter} simulation, the broad region of upstream lower effective pressures occurs in both simulations (e.g., compare the right panels of Fig. 3 **b–d** to the right panels of Supplementary Figure 8 **b–d**). The upstream spatial extent of lower effective pressures within the Helheim Glacier main, southern, and northern tributaries is roughly equivalent between the M_{229} and M_{winter} simulation. Though it is likely not useful to interpret the model output this finely, the absolute magnitude of the decrease in effective pressure 0.43 d after the start of the M_{winter} lake-drainage simulation at a location about ~ 5 km upstream from the location of the lake (right panel of Supplementary Fig. 8d) is greater than the magnitude of the decrease in effective pressure at the equivalent time and place in the M_{229} simulation (right panel of Fig. 3d).

Taken together, however, the upstream effects of the simulated lake drainage on modeled effective pressure estimates in the M_{229} and M_{winter} simulations do suggest a broad reduction in coupling between the glacier and its bed during these simulated lake-drainage events. We have added a clause noting this feature of the model results on L180–182.

Figure 3: Only the stations near the lake drainage appear to experience the increase in N following the lake drainage (on day 230) that is visible upstream and downstream of the lake location. Can this be explained in more detail? Are the stations near the terminus far enough removed that the pulse is dampened by the time it reaches them? Is the drainage system not “efficient” near the terminus? Is there sufficient water flowing in from the other branch to mute the influence of the lake drainage?

Figure 3: In panel **h**, it is difficult to distinguish all the different station lines, as some are stacked. It looks like only IS27, 28, and 29 experience the positive rebound in N after the conclusion of the lake drainage, which is the whole premise of this paper. So, what's going on at the bed around the lake compared to near the terminus? More detailed explanation or interpretation would be helpful.

Thank you for these two comments on Figure 3. In response, we have changed the line style in Fig. 3h (and Supplementary Fig. 8h) to improve visualization of N/N_{228} at the location of GPS stations that with a couple kilometers of each other, and thus show a similar modeled effective-pressure response. We also now include a new panel **i** inset within panel **h**. Panel **i** plots a narrow range of N/N_{228} (0.99–

1.01) from 2007/230.0–232.0 to better show the positive rebound in N that is also modeled at the locations of the near-terminus stations.

Given these changes to Fig. 3, it should now be more evident that the location of near-terminus GPS stations in the model domain does experience a positive (though modest) rebound in N following the lake drainage for the M_{229} simulation. For the M_{winter} simulation, it should now be more evident that the location of near-terminus GPS stations in the model domain returns to the pre-drainage values in N (Supplementary Fig. 8i), and that the location of lake-proximal GPS stations in the model domain has a smaller positive rebound in N following the lake drainage (Supplementary Fig. 8h) compared with the equivalent locations in the M_{229} simulation (Fig. 3h). This return of N to pre-drainage values at the location of all GPS stations in the model domain was previously indicated through the pulse-duration metric in the model as “the time between t_0 of N at station IS27 and t_{node} of N averaged across stations IS35–39, where t_0 of N is the time when $N/N_{228} < 0.98$ and t_{node} of N is the time when $N/N_{228} > 1.00$ following t_{peak} ” (L447–449); we hope the changes described here make it clearer.

That there is a difference in the magnitude of positive N/N_{228} ratios after the conclusion of the lake drainage between lake-proximal and near-terminus locations in the model domain is most likely a result of multiple factors. The drainage system is “efficient” near the terminus in the sense that flow is carried mainly through channels (but please see new Figure 4 and associated text in the Results section). The reviewer is correct that additional water flows in from other glacier tributaries such that the inputs from the lake drainage are a lower percentage of the total water flow near the terminus than they are at the lake. Additional factors important to consider here include (1) that surface melt production (a model forcing) is higher at lower elevations and (2) surface runoff inputs and basal melt inputs are integrated along the bed in the model domain as one moves from higher regions to the water outflow point modeled along the terminus (yellow circle in Figs. 3, 4, and S8), such that the terminus experiences a higher total water throughput than the region near the lake. A second factor for the difference in the magnitude of positive N/N_{228} ratios in the near-terminus region is that effective pressures in the near-terminus region prior to the simulated lake drainage are lower ($\sim 0.4\text{--}0.9$ MPa) than those in the lake-proximal region ($\sim 3.0\text{--}3.5$ MPa) (Fig. 3a). This is largely due to the glacier thickness and bed geometry in the near-terminus region resulting in the glacier being closer to flotation than in the tributary regions. Thus, the ocean boundary and the integration of surface melt from the tributary to the terminus region at Helheim Glacier likely both contribute to a smaller magnitude of positive N/N_{228} ratios after the conclusion of the lake drainage at near-terminus locations in the model domain. We have added a sentence explaining the above on L162–168.

Line 386: It would be helpful to describe what the basal melt rate represents and how it is calculated. (Geothermal flux and frictional heat from sliding? Although that should vary spatially, and it sounds like it is a uniform constant rate here). Is this basal melt applied everywhere in the model, or only to channel components?

We have revised the Methods subsection *Model domain and boundary conditions* to include these details. Basal melt rate m is calculated as the sum of melting produced by a spatially constant geothermal flux and melt generated from the frictional heat from sliding using a fixed basal sliding speed U_b of 500 m yr^{-1} and basal shear stress τ_b of 60 kPa over the model domain. Basal melt rate m is applied at every node in the model domain. Moreover, a small upstream basal melt flux is applied at the eastern boundary on the subglacial drainage system, equivalent to the basal melt rate (0.0262 m yr^{-1}) integrated over a 50-km-distance inland from the eastern boundary.

Basal melting increases the height of the cavity sheet layer (Hewitt (2013) Eq. 5a), which increases the discharge in the sheet (Hewitt (2013) Eq. 3), which results in an increasing melting rate of channel walls (Hewitt (2013) Eq. 6). Thus, basal melt is applied everywhere in the model.

Line 395: I am not convinced that $N=0$ is necessarily appropriate for the terminus. I believe Helheim is generally thought to be grounded at the terminus, but $N=0$ implies a floating terminus. It would be

helpful to explain why this choice was made, as opposed to setting the subglacial water pressure at the discharge equal to the overlying water pressure in the fjord.

We prescribed N to be zero at the marine-terminating margin of Helheim Glacier because, at the time of the lake drainage in 2007, the terminus position, glacier thickness, and glacier-bed elevation indicate that the terminus is nearly at flotation. Figure 1b of *Stevens et al. (2021)*, attached at the end of this response, shows a glacier cross-section along the main tributary of Helheim Glacier, using glacier surface and bed elevations from a 2008/212 Center for Remote Sensing of Ice Sheets (CReSIS) flight over the glacier. The solid vertical black line in both panels shows the terminus position on 2007/236, equivalent to the terminus position plotted in Figure 1a of the current study. The grey dashed line in Figure 1b of *Stevens et al. (2021)* below shows the glacier surface at which Helheim Glacier would be at flotation. As the CReSIS mapped glacier surface elevation (cyan line) and flotation elevation are nearly equivalent starting ~2 km inland of the glacier terminus on 2007/236, we chose to prescribe N to be zero at the glacier terminus.

Moreover, recent work from *Melton et al. (2022)*, who use observations of proglacial plume presence and absence at the surface of the fjord for a more recent time period (2016–2019) when the terminus is retreated back near its location in 2007, also support an interpretation of the Helheim Glacier terminus vacillating between being grounded and ungrounded over the melt season. This leads us to our overall description of Helheim Glacier as “lightly grounded” (*Kehrl et al., 2017*) and our choice to prescribe N to be zero at the glacier terminus. We have added a sentence explaining our choice, and a citation *Melton et al. (2022)*, in the Methods subsection *Model domain and boundary conditions*.

Lines 428-468: Permeability and englacial void ratio are prescribed as uniform over the entire domain. Finding the best fit values as done here may be helpful as a homogenized representation of what’s happening at the bed, but may be misleading by blurring the spatial heterogeneity.

The reviewer is correct that having a fixed englacial void ratio over the model domain does blur spatial heterogeneity in model output that would be related to a spatially variable englacial void ratio. Observationally constraining a spatially variable englacial void ratio across the model domain is not yet possible given our limited glacier-surface observations, so we choose to range across an “effective” value for the glacier system for this parameter.

For basal water sheet permeability, however, although there is a spatially uniform sheet permeability parameter (K_s) chosen for each model run, the effective hydraulic transmissivity of the cavity sheet layer varies in both space and time in response to the water input. In *Hewitt (2013)*, *Stevens et al. (2018)*, and here, this effective permeability is calculated as $K_s h^3$ (h is the height of the cavity sheet layer), and referred to as the hydraulic transmissivity T of the basal water sheet (Supplementary Figure 7 b,d). As the height of the cavity sheet layer h varies in space and time in response to water input, so too does the effective permeability of the basal water sheet. We have added a sentence to the Methods section **Subglacial hydrology model** to describe the spatiotemporal variability of effective hydraulic transmissivity in the cavity sheet layer.

Supplementary Figure 4: For how many days after the drainage event does the vertical lowering continue in IS27-29?

The three GPS stations that show vertical lowering following the drainage event were taken off the ice during DOY 235. We have revised Supplementary Figures 3 and 4 to include along-flow velocities and vertical station heights through the end of DOY 235. At this time, vertical lowering at IS29 appears to have stopped, but vertical lowering at stations IS27 and IS28 appears to be ongoing. Regrettably, we do not have a long-enough timeseries to determine the time of the end of the vertical lowering for stations IS27 and IS28.

Supplementary Table 1: It would be helpful to explain the choices for br and lr used in the model.

We assume the reviewer meant “ h_r ” and not “ b_r ”. A sentence explaining the choices of bed roughness height scale h_r and length scale l_r has been added to the Methods subsection *Model parameter space calibration*. Choices of bed roughness values here are the same as those used in *Banwell et al. (2016)* and *Stevens et al. (2018)*. Our primary justification for choosing these parameter values is a desire not to move too far outside of a region of model parameter space that we think we understand.

Figure 1 from Stevens et al. (2021): “**Helheim Glacier, East Greenland.** a) Location of (triangles) GPS stations and (circle) Automatic Weather Station (AWS). Orange triangle shows station on stagnant ice. Calving-front position (black dotted, solid, and dashed lines) shown on 4 July 2007 (DOY 185), 24 August 2007 (DOY 236), and 30 July 2008 (DOY 212). July 2007 velocities from the MEaSUREs Greenland Ice Sheet Velocity Map (Joughin et al., 2010; 2015) shown in grey contours at 1000 m yr⁻¹ intervals, with the 2000, 4000, and 6000 m yr⁻¹ contours labeled. Background is Landsat image from 1 July 2001 (DOY 182) acquired from the United States Geological Survey (<https://www.usgs.gov/>). Inset shows (star) location of Helheim Glacier in Greenland. Blue line on panel a shows 30 July 2008 (DOY 212) Center for Remote Sensing of Ice Sheets (CReSIS) flight line for ice-sheet surface and bed elevations shown in panel b (CReSIS, 2020).”

References cited in Response to Reviewer Comments

- Banwell A, Hewitt I, Willis I and Arnold N (2016) Moulin density controls drainage development beneath the Greenland Ice Sheet. *J. Geophys. Res. Earth Surf.* **121**, 2248–2269 (doi:10.1002/2015JF003801)
- Catania GA, Stearns LA, Moon TA, Enderlin EM and Jackson R (2020) Future Evolution of Greenland's Marine-Terminating Outlet Glaciers. *J. Geophys. Res. Earth Surf.* **125**, 1–28 (doi:10.1029/2018jf004873)
- Cowton T, Nienow P, Sole A, Wadham J, Lis G, Bartholomew I, Mair D and Chandler D (2013) Evolution of drainage system morphology at a land-terminating Greenlandic outlet glacier. *J. Geophys. Res. Earth Surf.* **118**(1), 29–41 (doi:10.1029/2012JF002540)
- de Juan J, Elósegui P, Nettles M, Larsen TB, Davis JL, Hamilton GS, Stearns LA, Andersen ML, Ekström G, Ahlstrøm AP, Stenseng L, Khan SA and Forsberg R (2010) Sudden increase in tidal response linked to calving and acceleration at a large Greenland outlet glacier. *Geophys. Res. Lett.* **37**(12), L12501 (doi:10.1029/2010GL043289)
- Hewitt IJ (2013) Seasonal changes in ice sheet motion due to melt water lubrication. *Earth Planet. Sci. Lett.* **371**, 16–25 (doi:10.1016/j.epsl.2013.04.022)
- Joughin I, Das SB, King MA, Smith BE, Howat IM and Moon T (2008) Seasonal Speedup Along the Western Flank of the Greenland Ice Sheet. *Science* **320**(5877), 781–783 (doi:10.1126/science.1153288)
- Hoffman M and Price S (2014) Feedbacks between coupled subglacial hydrology and glacier dynamics. *J. Geophys. Res. Earth Surf.* **119**(3), 414–436 (doi:10.1002/2013JF002943)
- Kehrl LM, Joughin I, Shean DE, Floricioiu D and Krieger L (2017) Seasonal and interannual variabilities in terminus position, glacier velocity, and surface elevation at Helheim and Kangerlussuaq Glaciers from 2008 to 2016. *J. Geophys. Res. Earth Surf.* **122**, 1635–1652 (doi:10.1002/2016JF004133)
- Melton SM, Alley RB, Anandakrishnan S, Parizek BR, Shahin MG, Stearns LA, LeWinter AL and Finnegan DC (2022) Meltwater drainage and iceberg calving observed in high-spatiotemporal resolution at Helheim Glacier, Greenland. *J. Glaciol.*, 1–17 (doi:10.1017/jog.2021.141)
- Nettles M, Larsen TB, Elósegui P, Hamilton GS, Stearns LA, Ahlstrøm AP, Davis JL, Andersen ML, de Juan J, Khan SA, Stenseng L, Ekström G and Forsberg R (2008) Step-wise changes in glacier flow speed coincide with calving and glacial earthquakes at Helheim Glacier, Greenland. *Geophys. Res. Lett.* **35**(24), L24503 (doi:10.1029/2008GL036127)
- Pimentel S and Flowers GE (2010) A numerical study of hydrologically driven glacier dynamics and subglacial flooding. *Proc. R. Soc. A Math. Phys. Eng. Sci.* **467**(2126), 537–558 (doi:10.1098/rspa.2010.0211)
- Schoof C (2010) Ice-sheet acceleration driven by melt supply variability. *Nature* **468**(7325), 803–6 (doi:10.1038/nature09618)
- Stevens LA, Hewitt IJ, Das SB and Behn MD (2018) Relationship Between Greenland Ice Sheet Surface Speed and Modeled Effective Pressure. *J. Geophys. Res. Earth Surf.* **123**(9), 2258–2278 (doi:10.1029/2017JF004581)
- Stevens LA, Nettles M, Davis JL, Creyts TC, Kingslake J, Ahlstrøm AP and Larsen TB (2021) Helheim Glacier diurnal velocity fluctuations driven by surface melt forcing. *J. Glaciol.*, 1–13 (doi:10.1017/jog.2021.74)